# Culturomics and Circular Agronomy: Two Sides of the Same Coin for the Design of a Tailored Biofertilizer for the Semi-Halophyte *Mesembryanthemum crystallinum*

**DOI:** 10.3390/plants12132545

**Published:** 2023-07-04

**Authors:** Eloísa Pajuelo, Noris J. Flores-Duarte, Salvadora Navarro-Torre, Ignacio D. Rodríguez-Llorente, Enrique Mateos-Naranjo, Susana Redondo-Gómez, José A. Carrasco López

**Affiliations:** 1Departamento de Microbiología y Parasitología, Facultad de Farmacia, Universidad de Sevilla, c/Profesor García González, 2, 41012 Sevilla, Spain; nflores@us.es (N.J.F.-D.); snavarro1@us.es (S.N.-T.); irodri@us.es (I.D.R.-L.); 2Departamento de Biología Vegetal y Ecología, Facultad de Biología, Universidad de Sevilla, c/Profesor García González, s/n., 41012 Sevilla, Spain; emana@us.es (E.M.-N.); susana@us.es (S.R.-G.)

**Keywords:** halophytes, PGPB, endophytes, bacterial consortia, biofertilizer, culturomics, plant-based culture medium, root colonization, metabolomics

## Abstract

According to the EU, the global consumption of biomass, fossil fuels, metals, and minerals is expected to double by 2050, while waste will increase by 70%. In this context, the Circular Economy Action Plan (CEAP) intends to integrate development and sustainability. In this regard, tailored biofertilizers based on plant growth-promoting bacteria (PGPB) can improve plant yield with fewer inputs. In our project, an autochthonous halophyte of the Andalusian marshes, namely *Mesembryanthemum crystallinum*, was selected for its interest as a source of pharmaceuticals and nutraceuticals. The aim of this work was to use a culturomics approach for the isolation of specific PGPB and endophytes able to promote plant growth and, eventually, modulate the metabolome of the plant. For this purpose, a specific culture medium based on *M. crystallinum* biomass, called Mesem Agar (MA), was elaborated. Bacteria of three compartments (rhizosphere soil, root endophytes, and shoot endophytes) were isolated on standard tryptone soy agar (TSA) and MA in order to obtain two independent collections. A higher number of bacteria were isolated on TSA than in MA (47 vs. 37). All the bacteria were identified, and although some of them were isolated in both media (*Pseudomonas*, *Bacillus*, *Priestia*, *Rosellomorea*, etc.), either medium allowed the isolation of specific members of the *M. crystallinum* microbiome such as *Leclercia*, *Curtobacterium*, *Pantoea*, *Lysinibacillus*, *Mesobacillus*, *Glutamicibacter*, etc. Plant growth-promoting properties and extracellular degrading activities of all the strains were determined, and distinct patterns were found in both media. The three best bacteria of each collection were selected in order to produce two different consortia, whose effects on seed germination, root colonization, plant growth and physiology, and metabolomics were analyzed. Additionally, the results of the plant metabolome revealed a differential accumulation of several primary and secondary metabolites with pharmaceutical properties. Overall, the results demonstrated the feasibility of using “low cost media” based on plant biomass to carry out a culturomics approach in order to isolate the most suitable bacteria for biofertilizers. In this way, a circular model is established in which bacteria help plants to grow, and, in turn, a medium based on plant wastes supports bacterial growth at low prices, which is the reason why this approach can be considered within the model of “circular agronomy”.

## 1. Introduction

Halophytes are salt-tolerant plants able to tolerate and thrive on soils containing high salt concentrations due to specific physiological and molecular mechanisms [1,2]. Their importance is being increasingly recognized due to the global acceleration of soil salinization and because they are sources of high-value products [3]. Some of them have been proposed as alternative crops with potential for ecological and economic applications, including renewable energy and ecological sustainability, environmental remediation, landscaping, and floriculture [4,5,6].

In addition, due to their ability to grow under stressful environmental conditions, halophytes can accumulate various secondary metabolites of interest (phenolic compounds, antioxidants, osmolytes, etc.) [7,8,9]. Particularly, they are taking on a relevant role as medicinal plants for their pharmaceutical and nutraceutical applications, and as functional foods and sources of unsaturated fatty acids [10,11,12].

*Mesembryanthemum crystallinum* L. is a moderate halophyte belonging to the Aizoaceae family. It is native to South Africa (Namibia), although it is currently established in the Mediterranean Basin, in Florida and Australia [13]. It is a resilient species that, besides growing on saline and poor soils, has minimal growth requirements and only a few infections [14,15]. This plant has had multiple traditional uses: the leaves and flowers are edible; it has had dermatological and cosmetic applications such as the manufacture of soaps and dyes. Regarding medicinal properties, historically, its fresh juice has been used to treat water retention and painful urination and to relieve lung inflammation [16]. Currently, it is arousing great interest as a source of bioproducts with high added value. It is considered a functional food due to the accumulation of polyols (pinitol and myo-inositol) that are used to prevent diabetes, treat obesity and prevent fatty liver disease [17,18]. It also has immunomodulatory properties in psoriasis treatment [19,20] and can be used in cosmetic creams as a moisturizer, having anti-aging and anti-wrinkle properties due to its antioxidant activity [21,22]. Finally, its antibacterial properties against *Escherichia coli*, *Pseudomonas aeruginosa*, and *Staphylococcus aureus* are documented [23].

The phytomicrobiomes of native halophytes are biotools to improve the environmental tolerance of these multifunctional plants [24,25]. A beneficial group of bacteria collectively called plant growth-promoting rhizobacteria (PGPR) deserves special attention. These bacteria, besides having biocontrol properties, benefit plants by promoting their growth and health, enhancing root system development, and improving their tolerance towards environmental stressors [26,27]. The growth-promoting effect is due to two types of mechanisms: direct and indirect. On the one hand, the production of auxins or siderophores by bacteria, N_2_ fixation, or phosphate/potassium solubilization are among the direct mechanisms, as well as the secretion of auxins. On the other hand, the indirect mechanisms include the induction of the systemic response, the secretion of extracellular enzymes (involved in the penetration of plant tissues and recycling organic matter), competition with pathogens in the rhizosphere, antibiosis, or the production of 1-aminocyclopropane-1-carboxylic acid (ACC) deaminase. In this regard, in response to a stressor, plants increase the production of the gaseous hormone ethylene from its direct precursor, 1-aminocyclopropane-1-carboxylic acid (ACC) [28]. Ethylene restricts plant growth by dampening the effect of auxins; hence, bacteria carrying ACC deaminase activity can decrease ethylene accumulation by ACC degradation, which translates into improved growth in inoculated plants [29]. The interest in *M. crystallinum*-associated PGPR has increased in recent years [30,31]. In these studies, several PGPR and endophytes which improved salt and heavy metal tolerance in the plant were isolated. Moreover, inoculation with these bacteria modulated the plant metabolome [32].

Despite the extensive knowledge generated on PGP microorganisms and the rising interest in their commercialization [29,33], the number of bacterial strains with PGP properties that are commercialized worldwide for agricultural applications is very scarce, not exceeding 25 strains belonging to 11 genera [34]. Hence, there is a need to expand the isolation, characterization, and selection of PGP strains with appropriate biological activities. Part of the problem is due to the difficulty of strain cultivation and the lack of inoculants’ stability and competitiveness in field applications [35].

Metagenomic analysis of microbiota detects mostly uncultured bacteria, but only culturable bacteria are useful as bioinoculants. Culturomics is a strategy recently applied to the isolation of fastidious gut microorganisms. The method allows describing the microbial composition by high-throughput culture [36]. It is based on the design of specific culture media adapted to particular microorganisms and selective and/or enrichment culture conditions coupled with MALDITOF-MS and 16S rDNA identification. This strategy allows the detection of minority populations, describing new species, and studying the interaction between different bacterial strains present in a given microbiota. Culturomics has allowed the isolation of gut microorganisms that could not have been isolated before [37]. A similar strategy has been applied to plant microbiomes [38,39].

In this work, we speculate on the possibility of preparing a new and specific culture medium based on *M. crystallinum* tissues as a strategy to isolate PGPR from the microbiome of this plant. The aim of this work is to evaluate a new culture medium based on plant extracts for the isolation of beneficial PGPRs and endophytes specifically associated with this particular host plant in order to design a biofertilizer capable of improving its growth and, possibly, modulating the accumulation of metabolites of interest.

## 2. Results

### 2.1. Soil Characterization

The characterization of the soil where a population of *Mesembryanthemum crystallinum* plants was found near the Guadiana River mouth in Ayamonte (SW Spain) was performed. Data from an initial evaluation of the soil properties are shown in Table 1. The soil sediment showed equivalent percentages of sand and silt and lower content of clay. The pH was around neutrality. The contents of organic matter and total nitrogen are in agreement with soil assays. Concerning the level of available phosphorous, it is relatively high in relation to the contents of organic matter and N, which suggests an accumulation of this element, probably due to the arrival of fertilizers to the river related to agricultural activity in this area. Regarding salinity, both the electric conductivity and the content of NaCl were moderate, according to an intertidal localization at the Guadiana River mouth. Furthermore, Ca concentration is relatively high, which can influence the establishment of particular plants. Regarding the soil concentration of heavy metals, all levels were far below the established thresholds by national and regional regulations; hence, no pollution of this soil could be considered [40].

### 2.2. Culturomics and Standard Approaches for the Isolation of Rhizosphere Bacteria and Endophytes from the Microbiome of Mesembryanthemum crystallinum

By using two different culture media, i.e., standard TSA and MA, the employed methodology allowed us to obtain two different collections of bacteria belonging to three different plant compartments, i.e., rhizosphere soil, roots, and shoots. A summary is shown in Table 2. In total, 47 isolates were obtained in the TSA medium: 10 from the soil, 26 as endophytes of the roots, and 11 as endophytes of shoots. Of these, 39 (83%) were Gram-positive, whereas 8 (17%) were Gram-negative.

A lower number of strains were isolated in Mesem Agar medium from all the compartments, namely 37; 11 of them were isolated from the rhizosphere soil; 15 as root endophytes; and 11 as endophytes of shoots. Of the 37 isolates, 34 (92%) were Gram-positive, whereas only 3 (8%) were Gram-negative. One remarkable fact was the isolation of only Gram-positive strains as root endophytes in both media.

Furthermore, the MA medium favored the isolation of Gram-positive strains, whereas a higher proportion of Gram-negative bacteria were isolated from rhizospheric soil upon using the TSA medium. Another difference between both media was the growth kinetics of the isolates; whereas on the TSA medium, most of the colonies appeared after 24–48 h, on the MA medium, the colonies developed more slowly, and the plates were finally incubated for 5 days in order to recover as much unique isolates as possible. Some steps in the preparation of the MA medium, as well as the aspect of some of the isolated bacteria on this medium, are shown in Appendix A.

All the bacteria isolated were characterized regarding PGP and enzymatic activities as described in Materials and Methods and identified by 16S rDNA sequencing. The complete list is shown in Appendix A, showing the genera and species isolated in each media, as well as the compartments they were recovered from. Most of the isolated strains (87%) were Gram-positive, while the rest (13%) were Gram-negative. The ratio of Gram-negative isolates was much higher on TSA than on MA, indicating that the last medium favored or selected Gram-positive isolates. Moreover, no Gram-negative bacteria were isolated from roots in either collection. Figure 1 shows Venn diagrams depicting the genera of bacteria isolated from rhizosphere soil (Figure 1A) as root endophytes (Figure 1B) and as shoot endophytes (Figure 1C). There were profound differences between the bacterial populations isolated in either media. The most commonly isolated genera from rhizosphere soil samples, *Bacillus* and *Pseudomonas*, were found in both media. Particularly, the following species were isolated: *B. subtilis*, *B. siamensis*, *B. velezensis*, and *P. chlororaphis*. On the TSA medium, three *Pantoea* strains were isolated: two strains of *P. agglomerans* and one strain of *P. alhagi.* In addition, one strain of *Glutamicibacter* was recovered in this medium. This genus was previously classified within *Arthrobacter*. Additionally, *Priestia megaterium* and *P. aryabhattai*, together with two strains of *Cytobacillus* and *Mesobacillus*, were isolated (Figure 1A).

Regarding root endophytes (Figure 1B), only Gram-positive bacteria were isolated. In both media, *Bacillus* (*B. cereus*, *B. amyloliquefaciens*, *B. subtilis*, *B. velezensis*, *B. pumilus*, *B. aryabhattai*, etc.) and *Rosellomorea* (*R. vietnamensis* and *R. aquimaris*) were isolated. *Rosellomorea* was formerly classified within the genus *Bacillus*. By contrast, each media allowed the isolation of specific genera: while *Cytobacillus* and *Mesobacillus* were isolated on TSA plates, *Microbacterium*, *Curtobacterium*, *Staphylococcus*, and *Priestia* were only isolated on MA plates.

In the case of shoot endophytes, both media allowed the isolation of strains of *Priestia* (*P. megaterium* and *P. aryabhattai*) and *Staphylococcus.* On TSA plates, strains of *Microbacterium* and *Lysinibacillus* were isolated, together with *Leclercia*, the only Enterobacteriaceae strain of the whole collection. On MA plates, *Curtobacterium* and *Pseudomonas* were found.

One noticeable result was the unique distribution of particular species in the different plant compartments. For instance, *Pseudomonas* could be isolated from rhizosphere soil and as endophytes of shoots, but neither *Pseudomonas* nor other Gram-negative strains were isolated from roots in any of the media.

### 2.3. Analysis of Plant Growth Promoting Properties and Extracellular Enzymatic Activities

Two sets of tests were performed on both collections of bacteria to characterize their PGP properties and extracellular enzymatic activities. The complete results of PGP properties are presented in Appendix A, whereas a summary of results is explained in Figure 2, showing that distinct populations were isolated in both media.

The most abundant PGP trait from TSA strain collection was nitrogen fixation (28% of strains harbored this capability), followed by siderophores formation (26%) and auxins synthesis (19%). The same trait pattern was found in plant compartments from which the microorganisms were isolated. A total of 71% of characterized strains had three or more out of six PGP traits. Strains isolated on MA medium had auxin synthesis (24.7%), nitrogen fixation (21.8%), and biofilm formation (20.8%) as the more abundant traits. This collection of strains had a different pattern of PGP traits by plant compartment. Auxin biosynthesis was the main trait of shoot endophytes (29.6%), biofilm formation (31%) and nitrogen fixation (24.4%) were the dominant traits for root endophytes, while phosphate solubilization (24%) alongside auxins biosynthesis (24%) were more represented among bacteria isolated from rhizospheric soil.

The TSA medium allowed the isolation of a higher number of strains from rhizosphere soil with nitrogen fixation, production of auxins, and secretion of siderophores (Figure 2A). Potassium solubilizers were only isolated on TSA plates. In contrast, the isolation of the highest number of phosphates solubilizers was achieved on the Mesem Agar medium. Among the root endophytes (Figure 2B), Mesem Agar led to the isolation of a higher percentage of strains with biofilm formation and siderophores production traits. In this case, no potassium solubilizers could be isolated on any of the media. Regarding shoot endophytes (Figure 2C), a higher number of strains able to form biofilms, fix nitrogen, or secrete siderophores were recovered on the TSA medium, whereas Mesem Agar was more suitable for the isolation of strains able to solubilize phosphate or secrete auxins. Again, the potassium solubilizers were rare, and only one strain was isolated on each of the media.

The complete results of extracellular degrading activities are listed in Appendix A, whereas a summary of results is shown in Figure 3. Again, it can be seen that distinct populations were isolated on both media, and in this case, the results were even more contrasting than those found for PGP properties. Regarding rhizosphere bacteria (Figure 3A), the TSA medium allowed the isolation of a higher percentage of strains with DNAase, cellulase, and pectinase activities, which are important for the degradation of plant cell walls, as well as for organic matter recycling. In contrast, MA led to the isolation of a higher ratio of strains with amylase and protease activities. Moreover, the only strain with lipase activity in both collections was isolated on MA plates. With regard to root endophytes (Figure 3B), both media led to similar results. However, MA allowed the isolation of strains able to degrade starch, and TSA permitted the isolation of strains with chitinase activity, which is an important trait for the degradation of fungal cell walls and presumably in biocontrol. Concerning shoot endophytes (Figure 3C), TSA led to the isolation of strains with pectinase and chitinase activities, whereas MA led to the isolation of a higher number of strains with amylase and cellulase activities. As a final summary of the results, Appendix A shows the percentage of traits by plant compartment and media.

### 2.4. Selection of the Consortia Members

The selection of the strains for each consortium was based on the following requirements: (a) the presence of rhizosphere bacteria and endophytes, particularly one member isolated from each compartment; (b) the concept of the “core genome”, i.e., the presence of as many as possible PGP properties and extracellular activities. Finally, the consortia were constituted by the following strains: S3, R5, and H4 for the TSA consortium and MS2, MR4, and MH8 for the MA consortium. Table 3 shows the identification of the strains and the properties of these bacteria. By combining the different properties of their members, the TSA consortium displayed the following properties: P-solubilization, biofilm formation, N_2_-fixation, auxin production, secretion of siderophores, K-solubilization, DNAase, amylase, cellulase, pectinase, and protease activities. In summary, it displayed 11 out of 13 desired activities. On the other hand, the MA consortium had the following properties: biofilm formation, N_2_-fixation, auxin production, siderophores production, phosphate and potassium solubilization, DNAase, amylase, cellulase, pectinase, and protease activities, i.e., 11 out of 13 desired activities. None of the consortia had lipase activity or chitinase activity.

### 2.5. Growth of the Selected Bacteria in Liquid TSB and Mesem Broth MB Liquid Media

In order to test the properties of Mesem Broth as a culture medium for bacteria, the growth curves of four selected strains (two rhizosphere and two endophyte bacteria from the TSA and MA collections) were followed in Mesem Broth and in the TSB medium (Figure 4). In addition, the selected bacteria were Gram-positive (R5 and MS2) and Gram-negative (S3 and MH8). The results showed that bacteria isolated on both the TSA and MA media were able to grow in both the TSB and MB liquid medium. Growth in TSB was always faster than that in MB. Furthermore, the lag phase was longer in MB as compared to that in TSB. For instance, for R5, the lag phase in TSB was 4 h, and in MB was 8 h. Finally, absorbances for bacteria grown for 48 h in TSB, a medium with a greater content of nutrients, were higher than in MB, where OD_600_ values were typically half when compared with those found in TSB.

### 2.6. Effect of Inoculation on the Germination of Mesembryanthemum crystallinum Seeds

Germination of seeds (non-inoculated and inoculated with either of the consortia) was followed for two weeks. The results are shown in Figure 5. Non-inoculated seeds displayed a moderate germination rate, reaching 50% in 3.5 days after sterilization and 68% after 7 days. Inoculation with either of the consortia improved this rate, reaching 50% in less than 3 days and full germination (100%) after 6 days. Moreover, no significant differences were registered between consortia.

### 2.7. Root Colonization

The formation of bacterial biofilms on roots is of the utmost importance for their colonization and for the display of plant growth-promoting activities due to the quorum necessary for displaying most of the PGP activities [41]. Root colonization by the two consortia was observed by low vacuum scanning electron microscopy (Figure 6). Different morphologies corresponding to cells of the three bacterial strains in each case can be observed. For instance, for the TSA consortium (Figure 6B), it is possible to observe coccobacilli in pairs corresponding to *Pantoea agglomerans*, together with long thin bacilli corresponding to *Pseudomonas chlororaphis* and big and thick bacilli of *Bacillus velezensis*. In the same way, three different morphologies were observed upon inoculation with the MA consortium (Figure 6C), specifically, big and thick bacilli of *Priestia megaterium*, shorter and thick bacilli of *Bacillus subtilis*, and long and thin bacilli corresponding to *Pseudomonas gessardii*. In this figure, the higher capacity for biofilm formation of the MA consortium can be observed with profuse root colonization. In order to test the individual capacity to form biofilms by the three members of the MA consortium, individual inoculations were performed. In Figure 6D, the bacilli of *Priestia megaterium* MS2 can be observed; some of them look thick in the central part of the cell as a consequence of the formation of the endospore. In addition, *Bacillus subtilis* MR4 was the bacterium with the highest colonization capacity (Figure 6E), forming large clusters of bacteria along the entire root sample, whereas *Pseudomonas gessardi* MH8 secreted a great amount of extracellular material, thus forming a dense biofilm where the bacteria were embedded (Figure 6F).

### 2.8. Effect of Inoculation with Bacterial Consortia on Plant´s Growth and Physiological Status

The inoculation with both consortia improved the growth of *M. crystallinum* in a similar way, with significant differences with control non-inoculated plants but without significant differences between both inoculants (Table 4, Figure 7). Shoot biomass was enhanced by 248% upon inoculation with both inoculants, whereas root biomass showed an increase of 272% with the TSA inoculum and 309% with the MA consortium. Concerning the effect on the physiological status of the plants, inoculation with either consortium did not affect the functionality of photosystem II, the maximum quantum efficiency of the photosystem II photochemistry (which correlates with the number of functional PSII reaction centers [42]), or the actual efficiency of the PSII.

### 2.9. Modulation of Metabolites Accumulation upon Inoculation with Selected Consortia

The possibility of modulating the concentration of functional metabolites upon inoculation of *M. crystallinum* with the different bacterial consortia was explored. Figure 8 shows the accumulation of amino acids (Figure 8A), organic acids (Figure 8B), sugars (Figure 8C), and other metabolites (Figure 8D) in non-inoculated and inoculated plants with both consortia. There was a deep alteration in the concentration of many amino acids. Many of them, including the non-proteogenic GABA (gamma-aminobutyric acid), Ala, Pro, Tyr, and Val, differentially decreased by inoculation with the TSA consortium, whereas Gln increased significantly. By contrast, Glu decreased by inoculation with both consortia. Finally, Asn only decreased by inoculation with the MA consortium up to almost undetectable levels.

In contrast to amino acids, whose concentrations usually decreased by inoculation with both consortia, the accumulation of organic acids usually increased by inoculation. In particular, malate and citrate underwent the highest enrichment. For instance, the content of malate increased by 25 and 23% upon inoculation with the TSA and MA inoculants, whereas the concentration of citrate increased by 52% and 120% in both conditions. Other acids, such as succinate and formic acid increased between 30 and 130%. In particular, inoculation with the TSA consortium differentially affected the accumulation of succinate and acetate (which increased significantly) and lactate, whose concentration diminished only upon inoculation with the TSA consortium.

Concerning sugars, both inoculants promoted the accumulation of glucose and fructose, although the TSA inoculation promoted a higher accumulation of these sugars by 70 and 100%, respectively. In contrast, the MA consortium very much enhanced the accumulation of sucrose by 300%. Noticeably, the TSA inoculant strongly lowered the accumulation of *myo*-inositol to 20% of the level found in non-inoculated plants.

The accumulation of other metabolites such as chlorogenate, choline, and trigonelline was stimulated by inoculation with the TSA consortium (from 1.2 to up to 3-fold in the case of trigonelline) and also with the MA consortium (1.4-fold in the case of choline and 2.6-fold in the case of trigonelline).

## 3. Discussion

Plant growth-promoting bacteria and endophytes (PGPB and PGPE) interact with plants and produce beneficial effects. In fact, these plant-associated microbiomes are considered biotechnology tools [24,43]. Metagenomics allows the identification of the microbiome members and their functional roles; however, some of these bacteria cannot be cultivated on standard media [38]. Thus, culturomics is a high throughput approach for the isolation of new members of the microbiota with possible biotechnological activities [38].

The approach followed in our work was the use of a plant-derived culture medium for the isolation of rhizospheric and endophytic bacteria. The utilization of plant-derived media is one of the characteristics of plant high high-throughput culture methods [38]. However, some other techniques employed in other culturomics studies, such as microcolony cultivation or identification by matrix-assisted laser desorption/ionization–time of flight mass spectrometry (MALDI-TOF-MS) [37], were not approached in our work. Instead, identification was performed by 16S RNA sequencing. Our approach led to the isolation of a lesser number of strains in MA as compared to TSA, and therefore is not probably a culturomics approach “*sensu stricto*”. However, even though the number of strains isolated on MA was lower, this medium allowed the isolation of particular and specific strains not very usually recovered from this plant. Similar approaches based on plant-derived media were reported for the isolation and culturability of lichen-associated bacteria [44] and rhizosphere/endophytes from cactus *Opuntia ficus-indica* and succulents *Aloe vera* and *Aloe arborescens* [45].

In our case, this approach allowed the isolation of distinct microorganisms in both media, namely, standard TSA and specific Mesem Agar. In this context, particular bacterial genera that are not usually isolated on standard media, for example, *Curtobacterium*, were isolated from roots and shoots of *M. crystallinum* in MA. *Curtobacterium* is a genus within the Actinomycetales that has been cataloged as a phytopathogen causing the bacterial wilt of legumes [46]. However, some other strains have PGP and biocontrol activities [47,48]. Other genera isolated on MA were *Cytobacillus* and *Mesobacillus*. These genera were recently separated from *Bacillus* in a new phylogeny of this genus [49].

By contrast, the medium TSA favored the isolation of some Gram-negative strains, such as the Enterobacteriaceae *Leclercia* and the Erwiniaceae *Pantoea*. A *Leclercia* strain with PGP properties has been described in tomatoes, where it ameliorated plant status under salinity stress by means of secondary metabolism modulation [50]. Additionally, several *Pantoea* strains were isolated from distinct plant compartments on TSA plates. *P. agglomerans* is known for its PGP properties, but it is classified within the security group 2 [51], so its use as a bioinoculant is forbidden by European regulations, whereas in other countries, the use of PGP strains is allowed. However, other species of the genus, such as *P. alhagi*, were isolated on TSA plates. This bacterium is described as a novel endophytic bacterium with the ability to improve growth and drought tolerance in wheat [52]. In addition, one species only isolated on TSA was *Glutamicibacter arilaitensis*. Some strains within the same genus display PGP properties and ACC deaminase activity being able to alleviate salt stress in rice plants [53].

Another conclusion obtained from our experiments is that *M. crystallinum* seems to have a great affinity for Gram-positive bacteria. However, when determining PGP properties and enzymatic activities, some Gram-negative strains, in particular *Pseudomonas* and *Pantoea*, displayed the highest number of these traits; therefore, they were chosen for the consortia. Likewise, some *Bacillus* species, such as *B. subtilis* and *Priestia megaterium*, together with *Priestia aryabhattai*, also displayed a high number of PGP and extracellular activities. Thus, these bacteria were selected for the TSA and MA consortia.

One important aim of our study was to establish whether the MB liquid or MA solid media were able to support the growth of the isolated bacteria. This question is relevant in the context of circular agronomy, since the medium MA is not only useful in a culturomics approach, but a solution to transform plant waste originating from industrial processes into a secondary product to make bacterial culture media. The MB medium is able to support the growth of bacteria, and it is a suitable and low-cost option for the development of specific inoculants that symbiotically interact with particular plants [38].

Once the TSA and the MA consortia were established, our results demonstrated that both consortia were able to promote seed germination and plant growth. Many examples of the positive effect of bacterial inoculation on seed germination are described [54]. This could be related to the existence of extracellular activities such as cellulase, pectinase, amylase, etc., involved in the degradation of the seed cover, improving its germination [55]. In addition, both consortia favored the growth of plants, particularly the MA one. The effect on growth does not seem to be related to the modulation of the efficiency of photosystem II, since we did not find an effect either on the maximum quantum efficiency of PSII photochemistry (F_v_/F_m_) or on the actual efficiency of photosystem II (ՓPSII). Therefore, there must be other factors or activities that influence growth, for example, the production of auxins, or the effect on nutrient mobilization (P or K solubilization), N_2_ fixation, etc. [24,26]. On the other hand, the presence of extracellular activities allows better adaptation to the environment, recycling nutrients, or the penetration of certain strains in plant tissues. For instance, cellulase, pectinase, and amylase allow the degradation of the plant wall and are important for the entry of endophytes [56].

For these activities taking place, the effective colonization of the root is necessary [57]. In fact, our study has shown the observation by low-vacuum SEM of the biofilms formed by the bacteria. It is noticeable that different patterns of colonization are observed: the strain *Bacillus subtilis* MR4 forms long chains occupying the depressed zones of the root surface, whereas *Pseudomonas gessardii* MH8 secreted a high amount of extracellular material for the formation of the matrix of the biofilm. In addition, *Priestia megaterium* is able to form spores that probably ensure the survival of the bacterium in the plant rhizosphere under unfavorable conditions [58].

An outstanding aspect of our results is the modulation of *M. crystallinum* metabolome, especially relevant in plants with pharmaceutical, nutraceutical, or cosmetic applications. It is noticeable that inoculation with both consortia decreased the concentration of many amino acids, particularly with the TSA consortia. The catabolism of amino acids is related to nitrogen recycling and ammonium transfer to other molecules, whereas the carbon skeletons usually are allocated to other tissues or feed the TCA cycle for energy production [59].

In contrast to the drop in amino acid content, the plant seems to have a tendency to accumulate organic acids (such as malate, citrate, succinate, and formate) and sugars in response to inoculation. The TSA consortium promoted the accumulation of glucose and fructose, whereas the MA inoculum induced the accumulation of sucrose. Additionally, inoculation with the TSA consortium drastically diminished the concentration of *myo*-inositol, a phosphorous derivative from glucose with important roles in phosphorous storage, cell wall biosynthesis, and the management of salt stress or osmoregulation in plants [58]. In this sense, it could be thought that the plants inoculated with the TSA consortium showed less stress. In a previous study, it was shown that inoculation with these consortia increased the concentration of malate and citrate in the presence of heavy metal stress [59]. Other studies also show that inoculation with endophytes modulates the metabolome of this plant [33,60]. On the other hand, the specific modulation of the secondary metabolism, precisely of compounds with pharmaceutical interest, was observed. Both consortia stimulated the accumulation of chlorogenate, choline, and trigonelline but at different ratios. Whereas inoculation with the TSA inoculum favored the accumulation of chlorogenate and trigonelline, inoculation with the MA consortium promoted a higher accumulation of choline. All these compounds have pharmaceutical applications. For instance, chlorogenic acid, a main component present in coffee, has a wide range of potential health benefits, including anti-diabetic, anti-carcinogenic, anti-inflammatory, and anti-obesity effects [61]. Choline is an essential nutrient present in some foods and available as a dietary supplement. It is required for the synthesis of two major phospholipids of cell membranes (phosphatidylcholine and sphingomyelin) and for the synthesis of the neurotransmitter acetylcholine [62]. Finally, trigonelline is a plant alkaloid with therapeutic potential for diabetes and the treatment of central nervous system diseases, since it has hypoglycemic, hypolipidemic, neuroprotective, antimigraine, and sedative effects [63].

## 4. Materials and Methods

### 4.1. Collection of Soil and Plant Samples

Sampling of plants and soil was carried out in February of the year 2022 in the town of Ayamonte (Huelva, southwestern Spain, coordinates: longitude 37.203498011869875, latitude −7.4066337565546374), where a population of *M. crystallinum* was located. Using sterile gloves, plant samples were taken, both aerial part and roots. Rhizosphere soil samples were also taken to a depth of 15 cm. All samples were placed in sterile plastic containers, refrigerated on ice, and transported to the laboratory. For the collection of *M. crystallinum* seeds, sampling was previously performed in October 2021 in the same area.

### 4.2. Soil Characterization

Conductivity was measured using a Crison-522 (Spain) conductivity meter, whereas redox potential and pH were determined with a Crison pH/mVp-506 (Spain) portable meter. Samples (200 g) were crushed with mortar and pestle and sieved through a 2 mm sieve, and soil texture (percentages of sand, silt, and clay) was determined in triplicate using the Bouyoucos hydrometer method [64]. The content of organic matter was measured by the chromic acid titration method [65], whereas total nitrogen was determined by the method of Dumas [66] using an element autoanalyzer CNS-Trumac (LECO). Available phosphorous was determined upon extraction with sodium bicarbonate according to the method of Olsen [67], followed by the determination of soluble P according to [68]. Exchangeable cations were determined upon treatment with 1N ammonium acetate [69], whereas microelements were extracted by DTPA (diethylenetriaminepentaacetic acid) [70], and their concentration was determined by ICP-OES using a spectrophotometer iCE 3500 AAS (Thermo Fisher Scientific Inc., NYSE: TMO, New York, NY, USA).

### 4.3. Preparation of Plant-Based Culture Medium for Culturomics

For 1 L of plant-based culture medium, called Mesem Broth, 15 gr of fresh plant material (both roots and shoots) were cut into small pieces, mixed with 200 mL of 50 mM phosphate buffer pH 7.0, and triturated with a mixer to obtain a homogeneous suspension. The volume was adjusted to 1 L with buffer. The medium was sterilized in autoclave for 20 min at 121 °C and 1 atm. The liquid medium was not fully transparent because small vegetal tissue rests were present in suspension, but this fact was not an impediment for subsequent use. For the preparation of solid medium called Mesem Agar, 15 gr of bacteriological agar (Difco) was added to 1 L of Mesem Broth before autoclaving.

### 4.4. Isolation and Identification of Rhizosphere Bacteria and Endophytes on Standard and Plant-Based Media

The isolation of rhizosphere bacteria was performed by preparing a suspension of 10 g rhizosphere soil in 50 mL of physiological solution (0.9% NaCl). The mix was vortexed for 5 min and let stand for 15 min. Serial dilutions from 10^−1^ to 10^−4^ were performed in physiological solution, and 100 µL of each dilution was stricken out on plates of either tryptone soya agar (TSA) or Mesem Agar. Plates were sealed and incubated at 28 °C for 24–48 h for the TSA medium and from 2 to 5 days in medium Mesem Agar (in our hands, the development of colonies on Mesem Agar was slower than that on TSA plates). Individual colonies with different apparent morphologies were further purified on TSA or Mesem Agar plates. Gram staining was performed to assess purity, and liquid cultures with 20% glycerol were stored frozen at −70 °C to establish two independent collections.

The isolation of bacteria from the rhizosphere was performed by preparing a suspension of 10 g of rhizospheric soil in 50 mL of physiological solution (0.9% NaCl). The mix was vortexed for 5 min and let stand for 15 min. Serial dilutions from 10^−1^ to 10^−4^ were carried out with physiological solution, and 100 µL of each dilution was stricken out on plates of either tryptone soya agar (TSA) or Mesem Agar. Plates were parafilm wrapped and incubated at 28 °C for 24–48 h for the TSA medium and from 2 to 5 days in Mesem Agar medium (in our hands, the development of colonies on Mesem Agar was slower than on TSA plates). Individual colonies with different apparent morphologies were further purified on TSA or Mesem Agar plates. Gram staining was performed to assess culture purity, and 20% glycerol liquid cultures stocks were stored at −70 °C, establishing two independent collections.

Identification of all strains was performed by 16S rDNA sequencing. For that, rapid genomic DNA extraction was performed by centrifuging 1 mL of overnight cultures in 1.5 mL polypropylene microcentrifuge tubes. Pellets were resuspended in 100 µL water and heated at 95 °C for 10 min and subsequently frozen at −80 °C for 10 min. After defrizzing, samples were centrifuged, and PCR was performed with 1 µL of the supernatant using the high fidelity Taq-Polymerase Q5 (New England Biolabs Inc., Ipswich, MA, USA) and standard primers for 16SrRNA (16S-27F, 16S-1488R/16S-1541R). PCR products were visualized by gel electrophoresis, and near full-length 16S RNA sequences were obtained by Sanger sequencing at the Catalunya Center of Technology Eurecat (Barcelona, Spain). Sequences were compared to the database to assess the identification of strains using BLAST and EzTaxon [71,72].

### 4.5. Analysis of Plant Growth Promoting Properties of the Isolates

The solubilization of phosphate was performed in NBRiP agar plates containing insoluble Ca_3_(PO_4_)_2_ as the sole P source [73]. After incubating the plates at 28 °C for 5–7 days, the appearance of a clear halo around the colonies was indicative of P solubilization. The diameter of the halo was measured as a semi-quantitative determination of P solubilizing ability. Analogously, the solubilization of K was performed in plates containing insoluble mica (potassium aluminum silicate) as the sole source of potassium [74]. After 5–7 days of incubation at 28 °C, the appearance of a clear halo around the colony indicated K solubilization, and the diameter of the halo was measured. Production of auxins was determined in liquid cultures grown in TSB medium containing 1 mM tryptophan. After 72 h incubation at 28 °C, the cultures were centrifuged, and 1 mL of Salkowsky reagent was added to 1 mL of the supernatants. The absorbance at 535 nm was determined, and the concentration of auxins was calculated against a standard curve made with commercial indoleacetic acid [75]. Siderophore production was assayed in plates of chromo-azurol-S (CAS) agar. Plates were incubated at 28 °C for 5–7 days, and the appearance of an orange halo around the colonies indicated the production of siderophores. The determination of the diameter of the halo was used as a semi-quantitative measure of siderophore production [76]. N_2_ fixation was assayed on nitrogen-free medium (NBF). Since some published media actually have significant concentrations of N in their composition (due to EDTA, hormones, vitamins, etc.), the composition of the medium used in our experiments, per liter, was as follows: 5 g malic acid, 0.5 g K_2_HPO_4_, 0.2 g MgSO_4_∙7H_2_O, 0.1 g NaCl, 0.02 g CaCl_2_, 1 mL of 0.1 M FeCl_3_, and 2 mL of trace elements containing, per liter, 0.2 g Na_2_MoO_4_∙2H_2_O, 0.235 g MnSO_4_∙H_2_O, 0.28 g of H_3_BO_3_, 0.008 g CuSO_4_∙5H_2_O, and 0.024 g ZnSO_4_∙7H_2_O. Final pH was adjusted to 6.8 with KOH, and for the solid medium, 15 g L^−1^ of bacteriological agar was added. The bacterial colonies were subsequently stricken thrice on this N-free medium, taking one individual colony previously grown on N-free medium and striking the colony again in N-free medium (3 consecutive replications). Only bacteria forming big colonies after three replications were considered N_2_ fixers.

The enzymatic degrading activities were assayed as follows [77]. Protease activity was assayed in plates containing casein, whereas lipase activity was assayed in plates containing Tween 80. Plates were incubated at 28 °C for 5–7 days. The bacteria were streaked as a line in the middle of the plate with an inoculation loop. The disappearance of the substrate (casein or Tween 80) produced a clear halo around the bacterial growth in positive strains. Amylase was assayed in plates containing Starch Agar (Difco). After incubation for 5–7 days at 28 °C, the disappearance of starch around the colonies was visualized by adding 10 mL of Lugol’s iodine on the surface of the medium. DNAase activity was performed on plates of DNA Agar medium (Difco) incubated at 28 °C for 72–96 h. After incubation, the disappearance of DNA around the strike was revealed by adding 10 mL of 10 mM HCl on the surface of the medium. Cellulase activity was assayed on plates containing carboxymethylcellulose. After incubating the plates at 28 °C for 5–7 days, the appearance of clear halos around the strikes was revealed by adding 1% Congo Red for 20 min and decolorization with 1 M NaCl for 5 min. Pectinase activity was tested in plates containing a mineral medium supplemented with pectin. Plates were incubated at 28 °C for 5–7 days. After incubating the plates at 28 °C for 5–7 days, the appearance of clear halos around the bacterial growth was revealed by flooding the plates with a solution of 1% CTAB (cetyltrimethyl ammonium bromide). Finally, chitinase activity was performed in plates containing colloidal chitin. As before, the disappearance of the substrate visualized as a clear halo around the strikes revealed the existence of chitinase activity.

### 4.6. Compatibility of Selected Strains and Formulation of Biofertilizers for Mesembryanthemum crystallinum

Two consortia of three bacterial strains jointly carrying most of the characterized PGP properties and enzymatic activities were selected independently from the TSA and MA collections. In order to test antagonistic effects among the selected strains, individual cultures of each of them were grown in liquid TSB medium for 24 h at 28 °C and 200 rpm. After incubation, optical density at 600 nm (OD_600nm_) of the cultures was adjusted to 1 with sterile TSB. A new culture was started by inoculating 3 mL sterile TSB with 20 µL of each culture. After 24 h of incubation, serial dilutions of the culture (up to 10^−9^) were prepared, stricken out on TSA plates, and incubated for 24 h at 28 °C. The presence of three different colonies with distinct morphologies confirmed the growth compatibility of the strains.

For the preparation of the inoculums, each bacterial strain was grown independently in 150 mL of liquid TSB for 24 h at 28 °C and 200 rpm. Cultures were centrifuged at 8000 rpm for 5 min, washed three times with sterile distilled water, and finally resuspended in 50 mL of sterile distilled water. Each inoculum was prepared by mixing equal volumes of the three selected cultures, and 5 mL of these mixtures were inoculated per pot containing one individual plant.

### 4.7. Effect of Inoculation on Seed Germination

Seeds of *M. crystallinum* (three batches of one hundred seeds each) were surface disinfected by treatment with 70% ethanol (2 min), washing with sterile distilled water, followed by treatment with 2.5% sodium hypochlorite (10 min) with gentle shaking, and finally exhaustive washing with sterile distilled water [78]. Once sterilized, they were deposited on moist sterile perlite in Petri dishes (25 seeds per plate). One hundred seeds (4 plates with 25 seeds each) were kept without inoculation treatment, whereas two other groups of 4 plates (one hundred seeds each) were inoculated with the TSA or the MA consortia. Plates were parafilm wrapped and incubated at 22 °C in the dark, and seed germination was followed for 14 days. During this period, 5 mL of sterile distiller water was added once a week.

### 4.8. Observation of Root Colonization by Low Vacuum Scanning Electron Microscopy

Roots from *M. crystallinum* inoculated seeds were collected from the germination plates to observe bacterial colonization by a low vacuum Phenom Pro Benchtop Scanning Electron Microscope (CITIUS, University of Seville). Small pieces of roots (2–4 mm) were cut with scalpel and placed directly on top of the microscope metal sampling device. Samples were inserted in the microscope and quickly frozen at −20 °C at low vacuum. The low vacuum scanning electron microscope allows direct observation of small organisms and living organs with high water content without pre-treatment in the micron and sub-micron range in a few minutes and with greater magnification than an optical microscope. Pictures of the root surface showing bacterial attachment and colonization were recorded, showing the scale of magnification.

### 4.9. Cultivation of Plants under Greenhouse Conditions

Seeds of *Mesembryanthemum crystallinum* (from the population found in Ayamonte, Huelva, SW Spain) were surface disinfected as described above. Disinfected seeds were pre-germinated on plates containing moist perlite in the dark at 20 °C for 48 h.

Seeds with emerging roots at the same developmental stage were first transferred to seedbeds for 1 month, and then to pots (pots of 0.5 volume; 1 plant per pot) containing a sterilized mixture of substrate/sand/perlite (8:1:1) in the greenhouse. One-third of the pots were watered once a week with 150 mL of tap water. Another third of the pots were watered as above and additionally inoculated once a month with the TSA consortium (5 mL per pot). Finally, the third group was watered every week as well and inoculated once a month with the consortium isolated in Mesem Agar (MA consortium). Plants were grown for four months in the springtime season, from the beginning of March to the end of June 2022. The greenhouse had controlled light and temperature conditions; when necessary, natural light was supplemented by fluorescent/incandescent lamps set to a photoperiod of 16 h of light to 8 h of dark; and the temperature was adjusted to 25 °C during the day and 15 °C during the night.

### 4.10. Evaluation of the Growth and Physiological Status of Plants

The physiological state of four-month-old plants (before the onset of flowering) was analyzed. The determinations were carried out at midday in fully expanded leaves (n = 7 determinations in 7 different plants) using a portable modulated fluorimeter (FMS-2, Hansatech Instrument Ltd., Pentney, UK). For that, the maximum quantum efficiency of the photosystem II photochemistry (F_v_/F_m_) and the actual quantum efficiency of PSII (ΦPSII) were calculated as follows. First, for the determination of the minimal fluorescence level (F_0_), leaf surfaces of 1 cm were adapted to dark for a period of 30 min using leaf clips. After this period, F_0_ was measured by first exposing the leaf surface to a pulse of <0.05 µmol∙m^−2^∙s^−1^ for 1.8 µs (too low to produce effects on plant photochemistry). The value of F_0_ was the average of values computed for a total period of 1.6 s. For the determination of the maximal fluorescence (F_m_), the same leaf surfaces were exposed to a pulse of saturating actinic light (15,000 µmol∙m^−2^∙s^−1^) for 0.7 s. The value of F_m_ was the maximum between two consecutive points. Then, F_v_ = F_m_ − F_0_ was calculated from F_0_ and F_m,_ and finally, the maximum quantum efficiency of PSII photochemistry (F_v_/F_m_) was obtained. According to [42], the ratio (F_m_ − F_0_)/F_m_ correlates with the number of functional PSII reaction centers.

The determination of the quantum efficiency of PSII (ΦPSII) was performed in the same leaf sections after adapting the plants to natural light for 30 min (F_s_). Then, a saturating pulse of actinic light (15,000 µmol∙m^−2^∙s^−1^) was applied for 0.7 s. to obtain the value of F_m_’ by temporary inhibition of PSII. The quantum efficiency of PSII (ΦPSII) was calculated from the formula (F_m_′ − F_s_)/F_m_′ [42].

At the end of the experiment, plant growth was evaluated. Plants were harvested and rinsed with tap water, and roots and shoots were separated. Total length and fresh weight of shoots and roots were measured. Part of the roots and shoots biomass was dried at 50 °C for 4 days until reaching a constant weight for dry mass determination. The rest of the biomass was quickly frozen in liquid N_2_ for metabolomics analyses after lyophilization, as described later.

### 4.11. Effect of Inoculation on the Accumulation of Metabolites in Shoots

Metabolomics analyses were performed at the Center for Edaphology and Applied Biology of Segura CEBAS-CSIC (Murcia, Spain). Frozen shoots were lyophilized using a freeze dryer Lyoquest −85 plus (Telstar), and samples of 50 mg were sent for metabolomics analyses (in triplicate). The concentrations of amino acids, organic acids, sugars, and other metabolites were determined following an untargeted analytical approach by ultra-high performance liquid chromatography coupled to a quadrupole time-of-flight mass spectrometer (UPLC-QtoF-MS/MS) [79].

### 4.12. Statistical Analysis

The program Statistica v. 6.0 (Statsoft Power Solutions, Inc., Hamburg, Germany) was used for statistical analysis. Plant biomass data are calculated as means of n = 10, whereas data of chlorophyll fluorescence are means of 7 determinations in 7 different plants. For its part, previously to compare between means, data were checked for normality with the Kolmogorov–Smirnov test and for homogeneity of variance with the Brown–Forsythe test. Then, comparison between means was performed by One-way ANOVA (F-test). In Tables and Figures, significant differences at *p* < 0.05 are indicated by asterisks.

## 5. Conclusions

Our results support the usefulness of a culturomics approach for the isolation, characterization, and selection of specific microorganisms from plant microbiota, as well as the design of effective microbial consortia capable of promoting plant growth and modulating plant metabolome to increase the content of high-value secondary metabolites with pharmaceutical or nutraceutical properties. In addition, our approach can be envisioned as a model of circular agronomy since it uses vegetal wastes/residues for the development of culture media supporting bacterial growth.

## Figures and Tables

**Figure 1 plants-12-02545-f001:**
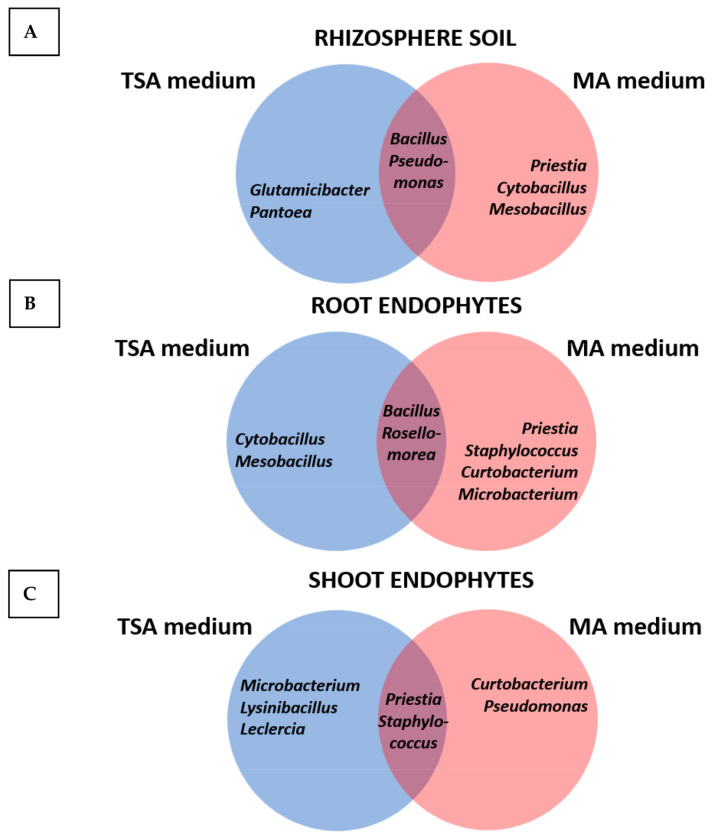
Venn diagrams showing the genera isolated from rhizosphere soil (**A**) as root endophytes (**B**) and as shoot endophytes (**C**) using either the TSA or the MA media.

**Figure 2 plants-12-02545-f002:**
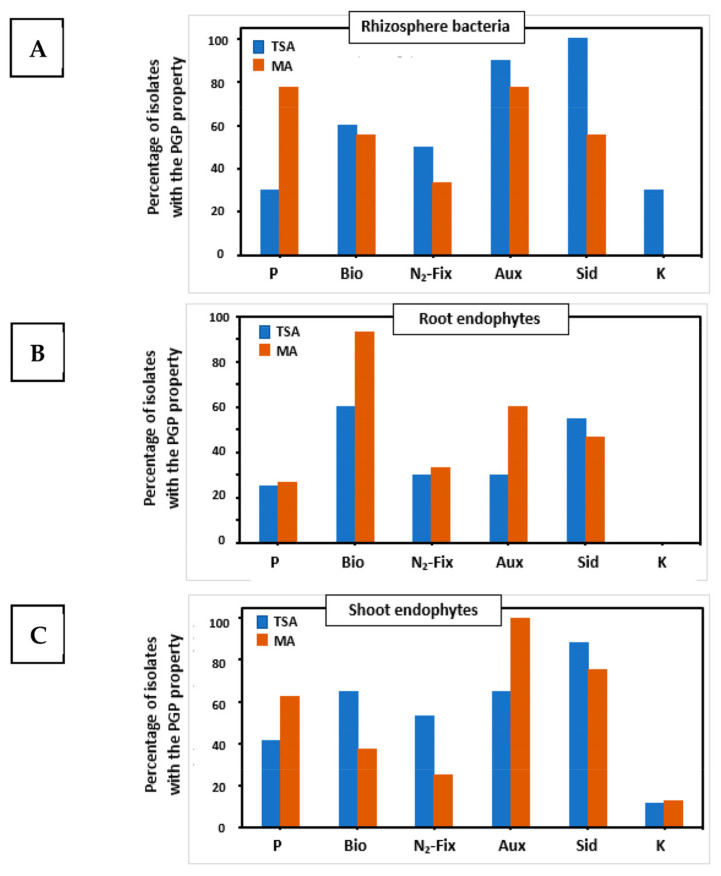
Percentages of strains with particular plant growth-promoting properties (PGP) isolated from the three compartments investigated using standard TSA medium (blue bars) or Mesem Agar (brown bars). Plant compartments: (**A**) rhizosphere soil; (**B**) roots; and (**C**) shoots. PGP properties: P: phosphate solubilization; Bio: formation of biofilms; N_2_-Fix: fixation of nitrogen; Aux: production of auxins; Sid: secretion of siderophores; and K: solubilization of potassium.

**Figure 3 plants-12-02545-f003:**
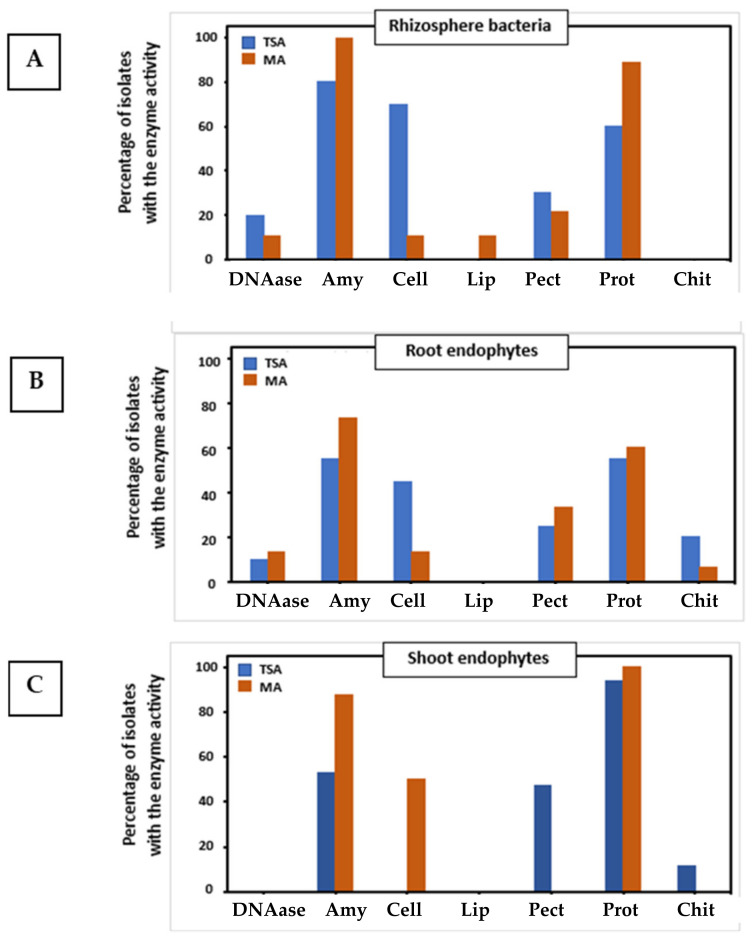
Percentages of strains with particular extracellular degrading activities isolated from the three compartments investigated using standard TSA medium (blue bars) or Mesem Agar (brown bars). Plant compartments: (**A**) rhizosphere soil; (**B**) roots; and (**C**) shoots. Degrading activities: DNAase; Amy: amylase; Cell: cellulase; Lip: lipase; Pect: pectinase; Prot: protease; Chit: chitinase.

**Figure 4 plants-12-02545-f004:**
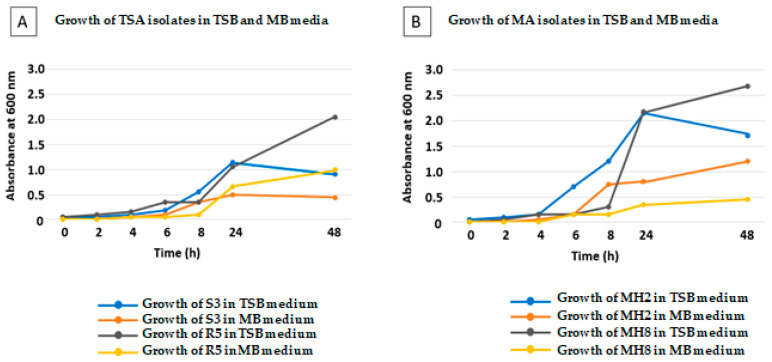
Growth curves of bacteria initially isolated in TSA medium (**A**) or in MA medium (**B**) in both TSB or MB liquid media. Two members of each consortium, one Gram-positive and one Gram-negative, were selected for the elaboration of the growth curves. In addition, there were one rhizosphere bacterium and one endophyte selected from each of the consortia.

**Figure 5 plants-12-02545-f005:**
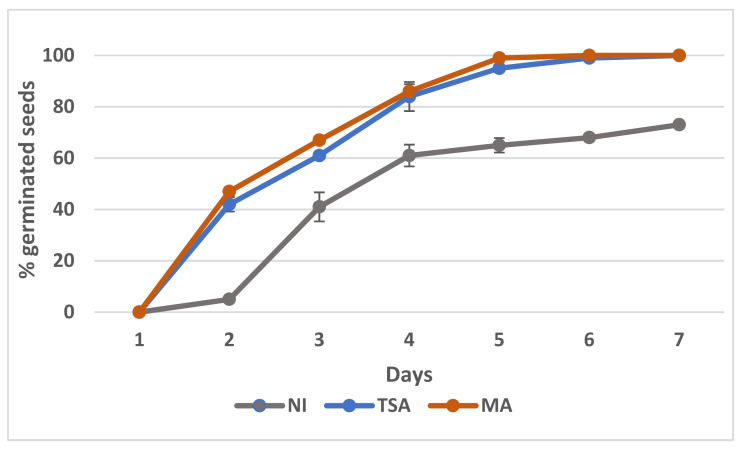
Germination curves of *Mesembryanthemum crystallinum* seeds without inoculation (NI) or inoculated with the TSA or MA consortia. Data are means of four determinations with 25 seeds each, and bars represent standard errors.

**Figure 6 plants-12-02545-f006:**
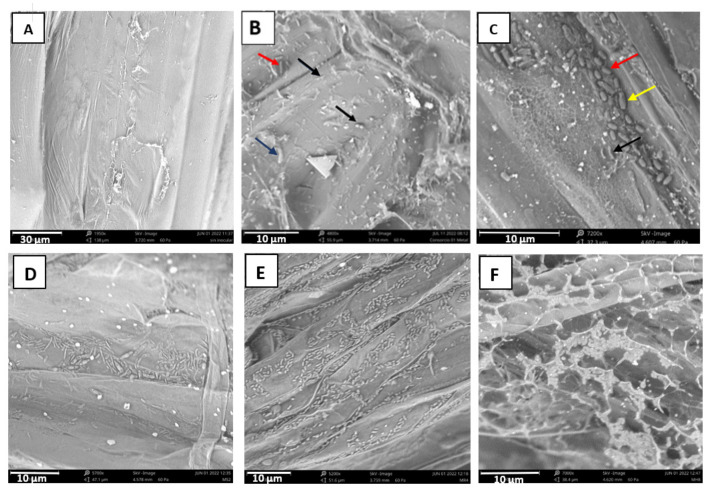
Colonization of *Mesembryanthemum crystallinum* roots by the TSA consortium (**B**) or the MA consortium (**C**). Arrows depict different bacterial morphologies in (**B**) (cocobacilli in pairs corresponding to *Pantoea agglomerans* (black arrow), together with long thin bacilli corresponding to *Pseudomonas chlororaphis* (red arrow) and big and thicker bacilli (dark blue arrow) of *Bacillus velezensis*), and (**C**) (big and thick bacilli (yellow arrow) of *Priestia megaterium*, shorter and thick bacilli (red arrow) of *Bacillus subtilis*, and long and thin bacilli (black arrow) corresponding to *Pseudomonas gessardii*). (**A**) Non-inoculated control. (**D**) Colonization by *Priestia megaterium* MS2 inoculated individually (some engrossed bacteria show the formation of the central spore) and (**E**) colonization by *Bacillus subtilis* MR4; note the profuse colonization by MR4. (**F**) Colonization by *Pseudomonas gessardii* MH8 inoculated individually. Note the extreme production of extracellular material for biofilm formation.

**Figure 7 plants-12-02545-f007:**
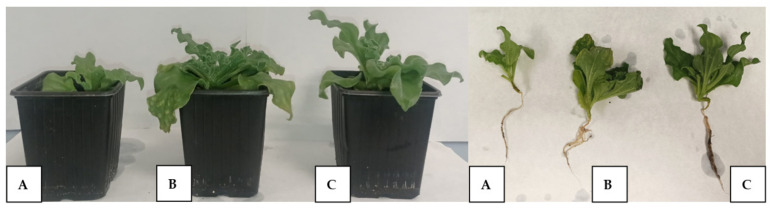
Comparative development of 4 months old *M. crystallinum* plants after inoculation with selected consortia. (**A**): non-inoculated, (**B**): inoculated with the TSA inoculant; and (**C**): inoculated with the MA consortium.

**Figure 8 plants-12-02545-f008:**
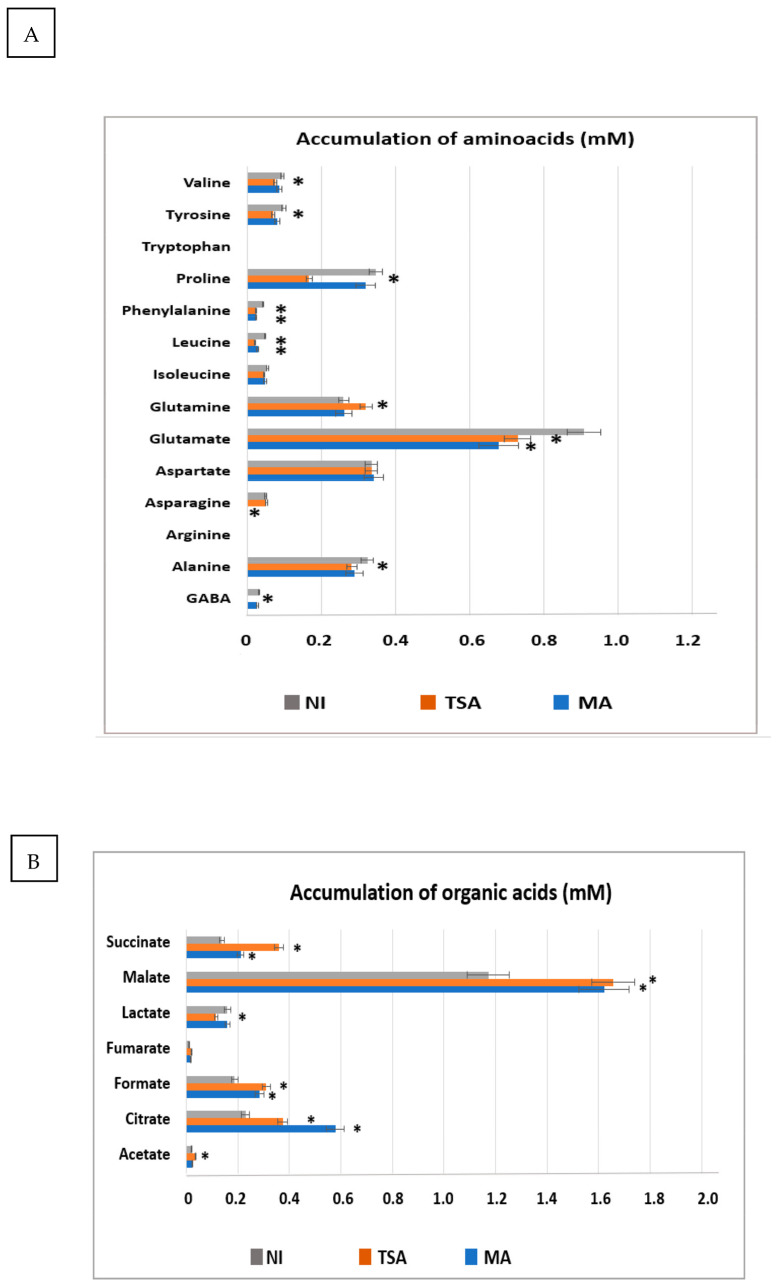
Modulation of metabolite accumulation in shoots of *Mesembryanthemum crystallinum* upon inoculation with TSA and MA consortia. (**A**): accumulation of amino acids in non-inoculated and TSA and MA inoculated plants. (**B**): accumulation of organic acids in non-inoculated and TSA and MA inoculated plants. (**C**): accumulation of sugars in non-inoculated and TSA and MA inoculated plants. (**D**): accumulation of other metabolites in non-inoculated and TSA and MA inoculated plants. Significant differences at *p* < 0.05 with regard to control non-inoculated plants are depicted with an asterisk.

**Table 1 plants-12-02545-t001:** Characteristics of soil where the population of *Mesembryanthemum crystallinum* was sampled. Data are means ± standard deviations of three determinations.

Soil Characteristics	
pH	7.24 ± 0.01
Electric conductivity (mS∙cm^−1^)	0.983 ± 0.027
Oxidable organic matter (%)	2.37 ± 0.07
Organic Carbon (%)	1.37 ± 0.04
Nitrogen Content (%)	0.26 ± 0.01
Available P (mg∙Kg^−1^)	49.70 ± 0.60
Content of macro- and micronutrients	
Ca (_c_mol∙Kg^−1^)	9.04 ± 0.57
Mg (_c_mol∙Kg^−1^)	3.60 ± 0.06
K (_c_mol∙Kg^−1^)	2.11 ± 0.07
Na (_c_mol∙Kg^−1^)	3.76 ± 0.11
Fe (mg∙Kg^−1^)	33. 93 ± 0.69
Mn (mg∙Kg^−1^)	18.14 ± 0.38
Zn (mg∙Kg^−1^)	20.97 ± 1.43
Cu (mg∙Kg^−1^)	5.21 ± 0.07
Cd (mg∙Kg^−1^)	ND
Pb (mg∙Kg^−1^)	ND
Texture	
Clay (%)	17.0 ± 1.8
Silt (%)	43.0 ± 1.0
Sand (%)	40.0 ± 1.4

ND, non-detected.

**Table 2 plants-12-02545-t002:** Summary of results obtained after the culturomics approach in medium Mesem Agar (MA) and standard medium (TSA).

Culture Medium	Compartment	Number of Isolates	Gram Staining	
			% Gram-Positive	% Gram-Negative
TSA				
	Rhizosphere soil	10	40 (4 strains)	60 (6 strains)
	Root	26	100 (26 strains)	0 (0 strains)
	Shoots	11	82 (9 strains)	18 (2 strains)
Mesem Agar				
	Rhizosphere soil	11	82 (9 strains)	18 (2 strains)
	Root	15	100 (16 strains)	0 (0 strains)
	Shoots	11	91 (10 strains)	9 (1 strain)

**Table 3 plants-12-02545-t003:** Identification of the strains selected for the TSA and MA consortia and the plant growth-promoting properties and extracellular activities of these bacteria. For some activities, only qualitative determination was performed, expressed as − (negative) or + (in particular, the symbols +, ++, and +++ were used for expressing the degree of activity). In other cases, such as phosphate and potassium solubilization, or auxin production, a quantitative determination was performed. In these cases, data are mean of three determinations.

Isolation Medium	TSA			MA		
Strain	S3	R5	H4	MS2	MR4	MH8
Identification	*Pantoea* *agglomerans*	*Bacillus* *velezensis*	*Pseudomonas* *chlororaphis*	*Priestia* *megaterium*	*Bacillus* *subtilis*	*Pseudomonas gessardii*
PGP properties						
P solubilization (Ø)	+ (0.3 cm)	−	+ (0.5 cm)	+ (0.3 cm)	−	+ (0.3 cm)
Biofilm formation (A_575nm_)	+ (0.150)	+ (0.156)	−	+ (0.122)	++ (0.463)	−
N_2_-fixation	+	+	−	−	+	−
Auxin production (µg mL^−1^)	+++ (17.62 µg mL^−1^)	−	++ (2.03 µg mL^−1^)	++ (2.0 µg mL^−1^)	−	++ (13.15 µg mL^−1^)
Siderophores production (Ø)	++ (1.4 cm)	−	+ (1 cm)	++ (1.5 cm)	+ (1.0 cm)	+++ (1.9 cm)
K solubilization (Ø)	−	−	+ (0.3 mm)	−	−	++ (0.6 cm)
Extracellular enzymes						
DNAase	−	+	−	−	+	−
Amylase	−	+	+	+	+	+
Cellulase	+	+	−	−	−	−
Lipase	−	−	−	−	−	−
Pectinase	−	+++	++	−	+++	−
Protease	−	+	+	+	+	+
Chitinase	−	−	−	−	−	−

**Table 4 plants-12-02545-t004:** Effect of inoculation with the TSA and MA inoculants on *Mesembryanthemum crystallinum* plant growth and physiological status. Data are means ± standard deviations of 10 determinations (biomass) and 7 determinations (fluorescence of the chlorophyll). Significant differences at *p* < 0.05 with regard to control non-inoculated plants are represented with different letters (a, b, c).

Parameter	Non-Inoculated	TSA	MA
Shoot fresh weight (g)	6.57 ± 3.39 ^(a)^	16.32± 7.67 ^(b)^	16.35 ± 1.68 ^(b)^
Shoot dry matter (g)	0.11 ± 0.03 ^(a)^	0.30 ± 0.10 ^(b)^	0.34 ± 0.02 ^(b)^
Root fresh weight (g)	0.13 ± 0.08 ^(a)^	0.26 ± 0.04 ^(b)^	0.36 ± 0.05 ^(c)^
Root dry matter (g)	0.012 ± 0.005 ^(a)^	0.024 ± 0.004^(b)^	0.034 ± 0.010 ^(c)^
Maximum quantum efficiency of PSII photochemistry (F_v_/F_m_)	0.75 ± 0.03 ^(a)^	0.76 ± 0.06 ^(a)^	0.71 ± 0.02 ^(a)^
Actual quantum efficiency of PSII (ɸPSII)	0.37 ± 0.02 ^(a)^	0.39 ± 0.04 ^(a)^	0.37 ± 0.03 ^(a)^

## Data Availability

Data are available in the text and in Appendix A.

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
