# Peer review of "Culturomics and Circular Agronomy: Two Sides of the Same Coin for the Design of a Tailored Biofertilizer for the Semi-Halophyte Mesembryanthemum crystallinum"

_plants, 2023, doi:10.3390/plants12132545_

Round 1
Reviewer 1 Report
The paper describes the application of consortium of isolates with PGP attributes, from two different culture media, and its application on growth of Mesembryanthemum crystallinum. The data presented here may be useful but the way it has been claimed to be a study of SynComs, and culturomics is not justified. With these terms, to be justified, the following criteria should be complied with.
Experiment, analyse and describe the microbiome of M. crystallinum. Provide information of its core microbiome through this experimental analysis.
Prove that the two respective consortia are in themselves, truly represents the functions of core microbiome – to be rightfully termed as Syncom.
Prove that the isolates which has been obtained were basically unculturable through previous studies/data of microbiome and had been cultured in this study – to be rightfully termed as culturomics. Also, the plant based media received lesser number of colonies, with slower growth and biased for G+ve organism. How possibly this could be a good tool for culturomics? This need to be explained too.
Otherwise, explain the work based on your data only, as isolation of PGP bacteria on two different media, characterization of their PGP attributes and application of two different consortium to obtain enhanced growth.
Also, delete pharmaceutical producing halophyte from the title, as the paper does not describe any such experiment undertaken here. The plant properties may be explained in introduction section, with support of literature.
Unfortunately, without these experimental justifications, it would not be wise to claim this as a culturomics and SynCom study.
Author Response
REVIEWER # 1
Dear Reviewer,
Thank you very much for the critical reading of our work and for your comments, which very much improve the quality of the work. We have carefully read all of them and elaborated an itemized response to each of them.
We have highlighted in yellow the corresponding changes done in the original submission, whereas the responses to reviewer #2 are highlighted in green. Reviewer # 3 had not asked for additional changes. We hope that the responses and changes done in the manuscript could satisfy the comments of the reviewer and that finally our work can be published in Plants.
- The paper describes the application of consortium of isolates with PGP attributes, from two different culture media, and its application on growth of Mesembryanthemum crystallinum. The data presented here may be useful but the way it has been claimed to be a study of SynComs, and culturomics is not justified. With these terms, to be justified, the following criteria should be complied with.
Thank you very much for your comments. We will try to defend some of our positions, but in other cases, we agree with the criticism of the reviewer and we have changed the terminology.
- Experiment, analyse and describe the microbiome of M. crystallinum. Provide information of its core microbiome through this experimental analysis.
Thank you very much for the comment. In fact, we also believe in the utility of this information in order to know the microbiome of the plant, and also what strains are specifically isolated on either media. However, this information has been already sent for publication to another journal in the context of a different study involving the effect of different bacterial inoculants on salt/drought tolerance in M. crystallinum.
- Prove that the two respective consortia are in themselves, truly represents the functions of core microbiome – to be rightfully termed as Syncom.
In this point, we agree with the reviewer that not all the tested properties are present in the bacterial consortiums. In fact, both consortiums had 11 of the 13 desired activities, but none of the members displayed lipase or chitinase activities. In this regard, we follow the recommendation of the reviewer and we have removed the term SynCom and we have used consortiums/inoculants through the entire manuscript.
Prove that the isolates which has been obtained were basically unculturable through previous studies/data of microbiome and had been cultured in this study – to be rightfully termed as culturomics. Also, the plant based media received lesser number of colonies, with slower growth and biased for G+ve organism. How possibly this could be a good tool for culturomics? This need to be explained too.
Otherwise, explain the work based on your data only, as isolation of PGP bacteria on two different media, characterization of their PGP attributes and application of two different consortium to obtain enhanced growth.
Thank you very much for the comment. However, in this point we disagree with the reviewer and we will try to fundament our response.
The aim of the work was not only using two different culture media (in that case we could have used, for instance, TSA and LB or whatever), but we did not intend to do that, but using a medium based on plant tissues and not in peptones, tryptone, meat extract, etc.). Probably media such as MA, based on plant extracts, are not so rich as standard TSA, LB… but the difference is that it contains all the metabolites of the plant and can “select” the bacteria more suitable for plant interaction. This is the reason for considering this medium as a tool for culturomics, in the sense of selecting the best strains for promoting plant growth.
We do not know whether all the bacteria isolated in MA medium are not cultivable, we have not tested the whole collection on TSA medium. Probably many of them are cultivable, for instance Pantoea, Leclercia… are Enterobacterales which are easy to cultivate. In spite of that, it is possible that some strains within the isolated genera could not be isolated in standard media, we do not know whether our particular strains could be cultivated on these media.
Moreover, the composition of the plant based medium leads to the isolation of particular strains that, in fact, were not isolated on TSA. Even though the medium is less rich, it allowed the selection of particular strains and this is why we have denominated this strategy as culturomics, since it is the use of a plant derived medium for the selection of the most suitable bacteria for interacting with this particular plant, which as the reviewer says, seems to prefer interacting with Gram positive strains.
Concerning the number of isolates, it is true that lesser bacteria were isolated on MA as compared to TSA. However, as we explained before, these bacteria are more suited for interaction with the host plant. In fact, in a previous work, we have used the two consortiums in metal polluted substrate using two different plants, namely, M. crystallinum and alfalfa (Flores-Duarte et al., 2023, cited in the references of the work). Surprisingly, our results demonstrated that the MA consortium worked better with its host plant M. crystallinum, whereas the standard TSA consortium worked better with alfalfa. We believe that these results support the culturomics approach, i.e., the specific plant derived medium “selected” the best strains for the particular host plant, evolved in the particular context of plant exudates in the rhizosphere or plant metabolites for endophytes.
- Also, delete pharmaceutical producing halophyte from the title, as the paper does not describe any such experiment undertaken here. The plant properties may be explained in introduction section, with support of literature.
Thank you for the comment. We have deleted the word “pharmaceutical” from the title. The properties of the plant are explained in the introduction. Furthermore, some secondary metabolites with pharmaceutical properties such as trigonelline, choline, etc., are investigated in this work. Anyway, we deleted “pharmaceutical” from the title.
- Unfortunately, without these experimental justifications, it would not be wise to claim this as a culturomics and SynCom study.
We have used “consortium” in the place of SynComs, however, we have kept the word culturomics as we explained before. We hope that our explanation could satisfy the questions of the reviewer.

Reviewer 2 Report
The manuscript by Pajuelo and colleagues present a new approach to isolate culturable bacteria specifically associated with a given host plant, namely an autochthonous halophyte, Mesembryanthemum crystallinum. This plant species has been selected for its interest as a source of pharmaceuticals and nutraceuticals. The new approach consisted of making and evaluating a new culture medium based on M. crystallinum extracts to obtain a “low-cost” medium for the isolation of beneficial PGPRs and endophytes specifically associated with this host plant. The ultimate goal of the authors was to design a biofertiliser capable of improving M. crystallinum growth and, possibly, able to modulate the accumulation of metabolites of interest.
I found this idea very clever and I congratulate the authors for their huge work. The experiments are sound and the results are presented clearly. I have only some minor comments that might improve the reading of the manuscript.
Comments
1/Editorial comment: my first comment is about English language because I found many mistakes regarding plural forms whose some are detailed below. Although these mistakes are minor, I think that the manuscript could benefit from a reading by an English-native speaker.
For example:
Line 97: Suppress the “s” in “strains cultivation”
Line 111-114: I would have written “The aim of this work is to evaluate a new culture medium based on plant extracts for the isolation of beneficial PGPRs and endophytes specifically associated with this particular host plant, in order to design a biofertiliser capable of improving its growth and, possibly, modulating the accumulation of metabolites of interest.”
Instead of : “The objective of this work is the assessment of a new plant extracts-based culture medium for the isolation of beneficial PGPR and endophytes specifically associated to this particular host plant, in order to design a biofertilizer able to enhance the its growth and, eventually, modulate the accumulation of metabolites of interest. “
Line 139: I would have written “In total, 47 isolates were obtained in TSA medium” rather than “A total of 47 isolates were obtained in TSA medium”
Line 807 : I would write “For that, rapid genomic DNA extraction was made by centrifuging 1 mL of overnight cultures in 1.5 ml polypropylene microcentrifuge tubes. “ instead of “For that, rapid mini- preps were made by centrifuging 1 mL of overnight cultures in 1.5 mL polypropylene microcentrifuge tubes” because doing miniprep is for plasmid extraction.
2/ comments about the results
Line 144: keep the same order to describe the bacteria isolated on MA medium: Gram+ first and then Gram-
Line 121: you wrote “However, the contents of N and P were much higher when compared to organic carbon, ….”
I don’t agree with this sentence …. Looking at Table 1, I can read:
Organic C content is 1.37 %, that is 1.37 g of C per 100 g of dry soil = 13.7 g of C per kg of dry soil;
Total N content is 0.26 %, that is 0.26 g of N per 100 g of dry soil = 2.6 g of N per kg of dry soil;
Total P content is 49.7 mg P per kg of dry soil = 0.0497 g P per kg of dry soil;
Hence, Organic C content is much higher than total N and P …. These values are in agreement with soil assays. Could the authors rewrite their text, please?
Also, I was confused when I read the line 771 (in Materials and Methods section) where it is written “available phosphorus” and not ”total P“ as in Table 1. For me, available P is extracted with the Olsen method for example. Could the authors precise which method they used to extract P?. Coming back to the results, if the P concentration given in Table 1 is Olsen P, a value of 49.7 mg P per kg of dry soil is quite high! And it could come from old fertilization practices, as suggested by the authors. But if it is total P content, this value is very low ….
As I wrote above, the text contains many mistakes regarding plural forms. Although these mistakes are minor, I think that the manuscript could benefit from a reading by an English-native speaker
Author Response
REVIEWER # 2
The manuscript by Pajuelo and colleagues present a new approach to isolate culturable bacteria specifically associated with a given host plant, namely an autochthonous halophyte, Mesembryanthemum crystallinum. This plant species has been selected for its interest as a source of pharmaceuticals and nutraceuticals. The new approach consisted of making and evaluating a new culture medium based on M. crystallinum extracts to obtain a “low-cost” medium for the isolation of beneficial PGPRs and endophytes specifically associated with this host plant. The ultimate goal of the authors was to design a biofertiliser capable of improving M. crystallinum growth and, possibly, able to modulate the accumulation of metabolites of interest.
I found this idea very clever and I congratulate the authors for their huge work. The experiments are sound and the results are presented clearly. I have only some minor comments that might improve the reading of the manuscript.
Dear Reviewer # 2,
Thank you very much for your comments and for the recognition of our work. We have read all your comments and elaborated an itemized response to them. The corresponding changes in the manuscript have been highlighted in green (the paragraphs highlighted in yellow correspond to responses to Reviewer # 1. For his part, Reviewer # 3 did not ask for any change). We hope that the modifications done in the document can accomplish the requirements as to be published in Plants.
Comments
1/Editorial comment: my first comment is about English language because I found many mistakes regarding plural forms whose some are detailed below. Although these mistakes are minor, I think that the manuscript could benefit from a reading by an English-native speaker.
The manuscript has been revised by an English native speaker. Changes are highlighted in green.
For example:
Line 97: Suppress the “s” in “strains cultivation”
The “s” has been deleted.
Line 111-114: I would have written “The aim of this work is to evaluate a new culture medium based on plant extracts for the isolation of beneficial PGPRs and endophytes specifically associated with this particular host plant, in order to design a biofertiliser capable of improving its growth and, possibly, modulating the accumulation of metabolites of interest.”
Instead of : “The objective of this work is the assessment of a new plant extracts-based culture medium for the isolation of beneficial PGPR and endophytes specifically associated to this particular host plant, in order to design a biofertilizer able to enhance the its growth and, eventually, modulate the accumulation of metabolites of interest. “
The paragraph has been rewritten. It is highlighted in green.
Line 139: I would have written “In total, 47 isolates were obtained in TSA medium” rather than “A total of 47 isolates were obtained in TSA medium”
Corrected.
Line 807 : I would write “For that, rapid genomic DNA extraction was made by centrifuging 1 mL of overnight cultures in 1.5 ml polypropylene microcentrifuge tubes. “ instead of “For that, rapid mini- preps were made by centrifuging 1 mL of overnight cultures in 1.5 mL polypropylene microcentrifuge tubes” because doing miniprep is for plasmid extraction.
Done.
2/ comments about the results
Line 144: keep the same order to describe the bacteria isolated on MA medium: Gram+ first and then Gram-
Done.
Line 121: you wrote “However, the contents of N and P were much higher when compared to organic carbon, ….”
I don’t agree with this sentence …. Looking at Table 1, I can read:
Organic C content is 1.37 %, that is 1.37 g of C per 100 g of dry soil = 13.7 g of C per kg of dry soil;
Total N content is 0.26 %, that is 0.26 g of N per 100 g of dry soil = 2.6 g of N per kg of dry soil;
Total P content is 49.7 mg P per kg of dry soil = 0.0497 g P per kg of dry soil;
Hence, Organic C content is much higher than total N and P …. These values are in agreement with soil assays. Could the authors rewrite their text, please?
The paragraph has been rewritten. We agree with the reviewer that total content of N and organic matter were in agreement with the C:N:P ratios for soils. So we have changed the text as : “The contents of organic matter and total nitrogen in agreement with soil assays. Con-cerning the level of available phosphorous is relatively high in relation to the contents of organic matter and N, which suggests an accumulation of this element probably due to the arrival of fertilizers to the river related to agricultural activity in this area.”
We kept the information about available P, as we explain in next question.
Also, I was confused when I read the line 771 (in Materials and Methods section) where it is written “available phosphorus” and not ”total P“ as in Table 1. For me, available P is extracted with the Olsen method for example. Could the authors precise which method they used to extract P?. Coming back to the results, if the P concentration given in Table 1 is Olsen P, a value of 49.7 mg P per kg of dry soil is quite high! And it could come from old fertilization practices, as suggested by the authors. But if it is total P content, this value is very low ….
We apologize for this mistake in Table 1. We refer to available phosphorous all the time. It has been determined by the method of Olsen, upon extraction with sodium bicarbonate. We have detailed all the procedures concerning soil analyses in the text now and we have included the corresponding references (from number 63 to number 68): “The content of organic matter was measured by the chromic acid titration method [63], whereas total nitrogen was determined by the method of Dumas [64] using an element autoanalyzer CNS-Trumac de LECO. Available phosphorous was determined upon extraction with sodium bicarbonate according to the method of Olsen [65], followed by the determination of soluble P according to [66]. Exchangeable cations were determined upon treatment with 1N ammonium acetate [67], whereas microelements were extracted by DTPA (diethylenetriaminepentaacetic acid) [68] and their concentration was determined by ICP-OES using a spectrophotometer iCE 3500 AAS (ThermoFisher Scientific)”.
Comments on the Quality of English Language
As I wrote above, the text contains many mistakes regarding plural forms. Although these mistakes are minor, I think that the manuscript could benefit from a reading by an English-native speaker.
The text has been revised by an English native speaker.

Reviewer 3 Report
The paper is interesting in a cognitive context and contributes relevant aspects concerning isolation, characterization and selection specific endophytic microorganisms as the promising alternative strategy supporting environmentally friendly approach and modulating plant metabolome to boost the content of secondery metabolites with diverse aptitudes. All the tables and figures are appropriate for this type of article. In general, the paper has a logical flow and it is refined in detail. The abstract well correspons with the main aspects of the work, methodology extensively described, results elaborately presented. The paper is very well written.
Author Response
REVIEWER # 3
The paper is interesting in a cognitive context and contributes relevant aspects concerning isolation, characterization and selection specific endophytic microorganisms as the promising alternative strategy supporting environmentally friendly approach and modulating plant metabolome to boost the content of secondary metabolites with diverse aptitudes. All the tables and figures are appropriate for this type of article. In general, the paper has a logical flow and it is refined in detail. The abstract well corresponds with the main aspects of the work, methodology extensively described, results elaborately presented. The paper is very well written.
Dear Reviewer # 3,
Thank you very much for your comments and for the recognition of our contribution.

Round 2
Reviewer 1 Report
Authors have addressed some of the concerns.
1. 'Consortiums', should be replaced with 'consortia'. Consortia are plural, consortium is singular. Correct accordingly.
2. The authors are willing to retain term 'culturomics' for the strategy used in this research. In my opinion, its their research; and therefore they should be allowed to if they find it that the culture conditions are sufficient to be termed as 'high-throughput culture methods', and they have achieved isolation of number of bacterial species, including otherwise belonging to microbial dark matter, to reach the real diversity of environmental samples. Some papers, that explains 'culturomics' are given below.
https://www.nature.com/articles/nmicrobiol2016203
https://www.sciencedirect.com/science/article/pii/S2090123219300803
https://www.nature.com/articles/s41579-018-0041-0
https://onlinelibrary.wiley.com/doi/10.1111/1469-0691.12032
https://www.ncbi.nlm.nih.gov/pmc/articles/PMC7719802/
The paper may be accepted after correcting the terms as given in point 1
Author Response
Dear Reviewer, thank you very much for your comments.
We have changed consortium by consortia through the whole document.
Moreover, we have introduced the following paragraph in the Discussion, indicating that our approach has some characteristics of the culturomics studies but we have not used some other methods such as cultivation of microcolonies, enrichment in specific media or selection of diverse incubation conditions
The approach followed in our work was the use of a plant-derived culture medium for isolation of rhizospheric and endophytic bacteria. The utilization of plant derived media is one of the characteristics of plant high-throughput culture methods [38]. However, some other techniques employed in other culturomics studies, such as microcolonies cultivation or identification by matrix-assisted laser desorption/ionization–time of flight-mass spectrometry (MALDI-TOF-MS) [37] were not approached in our work. Instead, identification was performed by 16S RNA sequencing. Our approach led to the isolation of a lesser number of strains in MA as compared to TSA, and therefore is not probably a culturomics approach sensu stricto. However, even though the number of strains isolated on MA was lower, this medium allowed the isolation of particular and specific strains not very usually recovered from this plant. Similar approaches based on plant derived media were reported for the isolation and culturability of lichen associated bacteria [44] and rhizosphere/endophytes from cactus Opuntia ficus-indica and succulents Aloe vera and Aloe arborescens [45].
We hope that our changes can be adequate for publication in Plants